# Evaluating Cultural Alignment in Multilingual LLM Translations

## 1. Abstract

This study explores the localisation capabilities of state-of-the-art multilingual large language models (LLMs) when translating figurative language, such as idioms and puns, from English into a diverse range of global languages. To investigate these challenges, this study evaluated a sample of 87 LLM-generated translations of e-commerce marketing emails across 3 leading models and 24 regional dialects of 20 languages. Human reviewers fluent in each target language provided quantitative ratings and qualitative feedback on faithfulness to the original's tone, meaning, and intended audience.

Findings suggest that, while leading models generally produce grammatically correct translations, culturally nuanced language remains a clear area for improvement, often requiring substantial human refinement. Notably, even high-resource global languages which top industry benchmarks, frequently mistranslated figurative expressions and wordplay; thus, challenging the assumption that data volume is the most reliable predictor of machine translation quality and introduces cultural resonance as a key determinant of multilingual LLM performance.

## 2. Introduction

Cultural nuance, especially figurative language such as wordplay, is central to effective human communication. This study investigates how well state-of-the-art multilingual LLMs handle figurative language when translating from English into a diverse range of global languages, addressing three core research questions:

(1) How reliably do LLMs translate idiomatic and figurative language across languages with varying resource availability and linguistic characteristics?

(2) Does linguistic proximity or shared structural features with English predict translation success?

(3) Are culturally nuanced translations more accurate in highly resourced, globally distributed languages than in smaller, regional ones?

## 3. Related Work

Recent research on multilingual LLMs reveals notable disparities in performance across high-resource and low-resource languages (Mujadia et al., 2025) and increasing scrutiny of the cultural and ethical implications of LLM-generated outputs (AlKhamissi et al., 2024; Li et al, 2024; Sterlie et al., 2024; Zhao et al., 2024). Existing literature supports the consensus that LLMs offer remarkable generative capabilities, but their real-world deployment across diverse linguistic contexts requires careful consideration of fairness, cultural sensitivity, ethical risk, and safety.

## 4. Methodology

### 4.1 Objective

This study investigates how well publicly available, leading large language models (LLMs) perform the task of translating and localising culturally nuanced language. The focus is on real-world use cases—specifically, scenarios in which marketing professionals with limited LLM expertise might rely on model outputs to localise copy from English to other languages. Marketing content often includes humour, cultural references, and idiomatic expressions, making it an ideal test case for evaluating multilingual performance and cross-cultural generalisation.

## 4.2 Materials

Three anonymised marketing emails were adapted from real commercial campaigns [Appendix 1]. These included seasonal and culturally specific references (e.g., Valentine's Day and Singles Day) and products related to food and body image – deliberately incorporating idiomatic language, such as humour and puns.

## 4.3 Participants

Twenty-two participants were recruited through convenience sampling for this pilot, covering 24 dialects across 20 languages (Appendix 2). All had prior experience with LLM-related projects and were fluent in English and at least one other language. Each participant evaluated translations into their language(s) of fluency.

## 4.4 Procedure

Participants received three anonymised outputs (one per model) for each email and asked to evaluate the outputs on four criteria, as follows:

- Content fidelity
- Tone fidelity
- Cultural and audience appropriateness
- Overall localisation quality

Each criterion was rated using a four-level scale, with "serious failures exist" on the low end and "very good or nearly perfect" on the high end (Appendix 3).

This methodology enables both quantitative comparison and qualitative insight into model performance across languages, content types, and localisation challenges.

## 5. Results

## 5.1 Overall Model Performance

The three LLMs tested demonstrated comparable performance; therefore, the data is presented anonymously to highlight cross-language patterns rather than comparative benchmarking. Across the entire dataset, localisation quality varied significantly by language, even for identical inputs processed by the same model. Appendix 4 shows the average localisation scores.

## 5.2 Cross-Linguistic Patterns

### *Language Family*

Languages more closely related to English generally achieved higher scores, though this relationship was inconsistent. Germanic languages performed well, supporting the hypothesis that lexical overlap with English contributes to translation quality. However, Romance languages showed mixed results.

Indo-Aryan languages showed strong overall performance. These results are consistent with prior research suggesting that LLMs perform more reliably in high-resource Indian languages like Hindi than in lower-resource regional languages like Urdu. Unexpectedly, Farsi, an Indo-Iranian language, achieved moderate scores despite known limitations in training data and significant linguistic distance from English (Abaskohi, 2024).

### *Orthography and Morphology*

While many high-performing languages used alphabetic systems, others with alphabetic scripts performed poorly. Languages with syllabary scripts performed exceptionally well.

Among orthographies, the logographic script demonstrated the lowest performance (Figure 1).

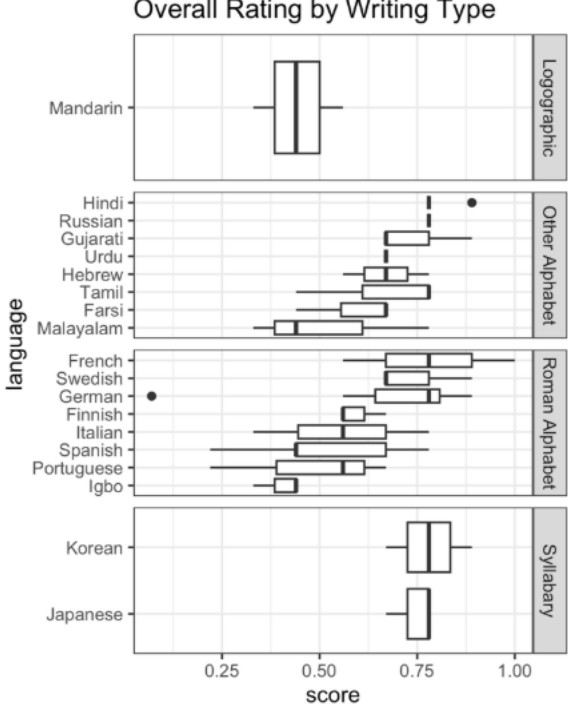

*Figure 1 Boxplot of overall ratings across languages grouped by writing system type*

Similarly, morphology did not demonstrate a direct relationship with translation quality. Agglutinative languages were overrepresented among top performers. Fusional languages also performed above the dataset's overall average. In contrast, isolating languages had the lowest scores.

*Resource Availability*

Languages with large speaker populations and global distribution, such as Spanish and Hindi, generally achieved higher scores. However, performance among medium-sized and regionally limited languages was mixed. Small, regional languages, such as Igbo, showed the weakest results overall. Mandarin further complicates this pattern, as it combines a large global speaker base with lower performance,

likely due to tokenization challenges and script complexity.

Notably, several smaller, regional languages exceeded the dataset's overall average (67.85%), challenging the assumption that resource availability—defined by speaker population size and global distribution—is a reliable predictor of LLM translation quality. Appendix 5 explores these outliers in greater detail.

## 6. Discussion

This study demonstrates that while large language models (LLMs) generate translations across a wide range of languages with minimal grammatical errors, overall localisation quality remains inconsistent. All languages required human correction to achieve natural-sounding translations, faithful to the tone and meaning of the original English. Resource availability, as measured by speaker population and global distribution, and orthography emerged as the greatest predictors of localisation quality.

### 6.1 Idiomatic and Figurative Language as Persistent Challenges

Evaluators consistently reported that while most LLM translations were grammatically correct, they often sounded unnatural or overly literal. Idiomatic expressions and puns were the most consistent sources of translation errors across languages. English wordplay such as "Will you brie mine?" was systematically mistranslated, resulting in awkward or confusing outputs that lost the intended humour or persuasiveness of the original.

The severity of these mistranslations varied across languages and was especially egregious in languages with significant structural and cultural distance from English, such as Mandarin, where direct translation of figurative expressions more frequently resulted in stiff or

confusing outputs. Appendix 6 presents four examples of mistranslations across languages as well as the evaluator's feedback.

## 6.2 Data Availability and Orthography Drive Performance

Model performance correlated most strongly with two factors: the availability of high-quality training data and the compatibility of the writing system with subword tokenisation methods. Linguistic proximity to English, by contrast, was not a reliable predictor of localisation quality.

## 6.3 Human Revision Remains Essential

Despite the generally high grammatical accuracy of LLM translations, evaluators rated 62.07% of outputs as requiring human revision. Qualitative feedback reported frequently needing to adjust word choice, mistranslated idioms, puns, and cultural references. Reviewers noted even technically correct translations often sounded awkward, overly formal, or culturally mismatched without human intervention.

## 7. Conclusion

This pilot demonstrates that while multilingual LLMs have achieved impressive levels of grammatical accuracy across diverse languages, they fall short when translating culturally nuanced language. Idioms, puns, and wordplay frequently resulted in literal, awkward, or contextually mismatched translations. These errors appeared across both high- and low-resource languages, challenging the notion that increased access to data equates to effective localisation. These findings suggest that current leading multilingual LLMs cannot yet be relied upon to consistently produce culturally aligned translations without expert human oversight.

## 8. Future Work

Phase two of this project is already underway which expands this methodology to include 2625 evaluations across seven models and fifteen languages.

## 9. Limitations

### Positivity of Feedback

Evaluator comments rarely used strongly negative terms like "inaccurate" or "inappropriate," which may indicate either generally acceptable translation quality or a bias toward constructive or positive framing.

### Language and Regional Representation

The dataset heavily represented Romance, Indo-Aryan, and Germanic languages—families with greater linguistic similarity to English.

### Scale of the Dataset

With 87 evaluated units and fewer than three evaluators per language, this research represents a small pilot study.

### Evaluator Expertise

While all evaluators were experienced in LLM evaluation tasks, their linguistic expertise and translation experience were not systematically controlled.

### LLM Model Diversity

The study anonymised model identities to reflect real-world user experience rather than comparative benchmarking.

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

## 12. Appendices

### *Appendix 1*

**Email 1**

Company: Sheffield's – a gourmet market in NYC

Subject: Will you brie mine? 🧀❤️🧀

Valentine's Day is almost here, and we've got the sweetest gift ideas for pickup or delivery throughout NYC.

Cheese Tasting Gift Boxes

This cheese lover's dream is thoughtfully assembled by our expert cheesemongers.  It all comes beautifully packaged in a keepsake tin, tied with a satin ribbon. Personalize it with a custom note on Sheffield's stationary.

Valentine's Charcuterie Boards Artfully displayed with the perfect accompaniments of fresh & dried fruit, nuts, honey, fig jam, espresso brownies, dark chocolate-covered strawberries, Valentine's candies, edible flowers and sliced baguette.

[order here]

We still have a limited number of handmade, chocolate-covered strawberries and floral arrangements available for pre-order! Give us a call today or stop by the shop before they're gone.

Wishing you a sweet Valentine's Day!

Sheffield's – Park Slope

Brooklyn, NY

**Email 2**

Company: Terra – an eco-friendly deodorant brand

Subject: This scent will transform your love life 😘✨

Swipe right this Single's Day

Use code: SINGLESDAY

[shop deodorant]

Hey [NAME]

This Singles' Day, give your deodorant game an irresistible upgrade – pair our newest reusable case design with a fragrance that's sure to make memories. Durable, stylish, compact, and zero waste.

Whether you're keeping yourself fresh for your partner, or looking to impress someone else, our new scents will leave a lasting impression.

MIX & MATCH OUR BEST-SELLING COMBOS

Lavender case x Tropical Paradise scent

Turquoise case x Orange Creamsicle scent

WHY TERRA?

Aluminum & paraben free. Zero-waste refills. 24-hour odor protection. All that in a case you'll be excited to reuse.

Terra Cosmetics

London N1C 4AB, United Kingdom

**Email 3**

Company: Muggable – an American novelty mug company

Subject: This Collection Has Us Feline Good 🐱

CAT'S MEOW

Our newest collection is the cat's pajamas, wait no – it's the cat's Mugs, Tumblers, Koozies, and Coasters!

[Shop Meow]

Rep your favorite feline at the office, on the go, and on your next Zoom call. Wait. Who are we kidding? They're already in all your Zoom calls.

© 2012 Muggable Inc. All Rights Reserved.

Los Angeles, CA, 90013, US

*Appendix 2*

Language resource availability was measured in this study by global speaker population (large = >200m, medium = 100m–200m, small = <100m) and geographic distribution (global = multiple regions, regional = limited geographic region).

*Summary of languages, regions, and resource levels included in this sample dataset. Future work will expand on the number of annotators and languages included in this pilot.*

| language | participant region(s) | size | global or regional |
|---|---|---|---|
| Farsi | Iran | small | regional |
| Finnish | Finland | small | regional |
| French | France, USA | medium | global |
| German | Germany, USA | medium | global |
| Gujarati | India | small | regional |
| Hebrew | Israel | small | regional |
| Hindi | India | large | global |
| Igbo | USA | small | regional |
| Italian | Italy | small | regional |
| Japanese | Japan | medium | regional |
| Korean | South Korea | small | regional |

| Malayalam | India | small | regional |
|---|---|---|---|
| Mandarin | China | large | global |
| Portuguese | Brazil | medium | global |
| Russian | Russia | medium | global |
| Spanish | Dominican Republic, Mexico, Spain, USA | large | global |
| Swedish | Sweden | small | regional |
| Tamil | India | small | regional |
| Urdu | Pakistan | medium | regional |

*Appendix 3*
*Count of annotations per language in the sample dataset*

| language | Number of evaluations |
|---|---|
| Farsi | 3 |
| Finnish | 3 |
| French | 9 |
| German | 12 |
| Gujarati | 3 |
| Hebrew | 3 |
| Hindi | 6 |
| Igbo | 3 |
| Italian | 3 |
| Japanese | 3 |
| Korean | 3 |
| Malayalam | 3 |
| Mandarin | 3 |
| Portuguese | 3 |
| Russian | 3 |
| Spanish | 15 |
| Swedish | 3 |
| Tamil | 3 |
| Urdu | 3 |
| Grand Total | 87 |

*Appendix 4*

*Average scores by language, region, and linguistic features* normalised to percentages based on the four-level evaluation criteria described above, grouped by language

| language | continent | language family | morphology | orthography | size | distribution | audience | meaning | tone | overall quality |
|---|---|---|---|---|---|---|---|---|---|---|
| Farsi | India | Indo-Aryan | Fusional | Abjad | small | regional | 70.37% | 55.56% | 77.78% | 59.26% |
| Finnish | Europe | Uralic | Agglutinative | Alphabetic | small | regional | 74.07% | 59.26% | 74.07% | 59.26% |
| French | Europe | Romance | Fusional | Alphabetic | medium | global | 79.01% | 74.07% | 76.54% | 77.78% |
| German | Europe | Germanic | Fusional | Alphabetic | medium | global | 75.93% | 67.59% | 77.78% | 69.44% |
| Gujarati | India | Indo-Aryan | Agglutinative | Abugida | small | regional | 66.67% | 77.78% | 66.67% | 74.07% |
| Hebrew | Europe | Semitic | Isolating | Abjad | small | regional | 66.67% | 59.26% | 62.96% | 66.67% |
| Hindi | India | Indo-Aryan | Fusional | Abugida | large | global | 79.63% | 79.63% | 75.93% | 79.63% |
| Igbo | Africa | Niger-Congo | Isolating | Alphabetic | small | regional | 40.74% | 37.04% | 37.04% | 40.74% |
| Italian | Europe | Romance | Fusional | Alphabetic | small | regional | 48.15% | 51.85% | 70.37% | 55.56% |
| Japanese | Asia | Japanese | Agglutinative | Syllabary | medium | regional | 92.59% | 77.78% | 96.30% | 74.07% |
| Korean | Asia | Korean | Agglutinative | Syllabary | small | regional | 92.59% | 85.19% | 88.89% | 77.78% |
| Malayalam | India | Dravidian | Agglutinative | Abugida | small | regional | 74.07% | 74.07% | 74.07% | 51.85% |
| Mandarin | Asia | Chinese | Isolating | Logographic | large | global | 48.15% | 44.44% | 51.85% | 44.44% |
| Portuguese | Europe | Romance | Fusional | Alphabetic | medium | global | 55.56% | 44.44% | 55.56% | 48.15% |
| Russian | Europe | Slavic | Fusional | Alphabetic | medium | global | 66.67% | 66.67% | 66.67% | 77.78% |
| Spanish | Europe | Romance | Fusional | Alphabetic | large | global | 61.48% | 60.74% | 65.93% | 53.33% |
| Swedish | Europe | Germanic | Fusional | Alphabetic | small | regional | 85.19% | 70.37% | 81.48% | 74.07% |
| Tamil | India | Dravidian | Agglutinative | Abugida | small | regional | 77.78% | 74.07% | 77.78% | 66.67% |
| Urdu | India | Indo-Aryan | Fusional | Abjad | medium | regional | 62.96% | 66.67% | 62.96% | 66.67% |
| Grand Total | | | | | | | 69.99% | 65.52% | 71.26% | 64.62% |

*Appendix 5*

*Positive Outliers Based on Language Size*

| Language | Average Across All Criteria | Interpretation |
|---|---|---|
| Korean | 86.11% | Strong model performance is likely supported by the tokenization advantages of its syllabary orthography and the accessibility of high-quality training data. |
| Swedish | 77.78% | Lexical overlap with English and a strong digital footprint are the most likely explanations for high translation quality, despite a small speaker base and regional use. |

| | | |
|---|---|---|
| Gujarati | 71.30% | Small but globally dispersed, Gujarati's strong performance suggests targeted investment by model builders in Indian language datasets. |
| Malayalam | 68.52% | Similar to Gujarati, these results indicate focused investment from model builders to optimise for regional Indian languages. |

*Appendix 6*

*Select Mistranslation Examples and Corresponding Evaluator Feedback*

| Language | Evaluator Feedback |
|---|---|
| Finnish | I changed the translation "keepsake tin" to "säilytettävään rasiaan" (instead of "muistorasiaan") because the word "muisto" can sometimes have an impression of something really gone, like forever, and it's not the best option here. |
| German | The models translated "cat's meow" as "katzenjammer" which has a negative/sad connotation in German, and I am very sure that a shop or company would definitely not want to use this to advertise for something. |
| Italian | *English and Italian are quite different languages, so puns have to be rewritten entirely. In this case, models #1 and #2 translated "will you brie mine?" as "Do you want to be my cheese?" which, albeit amusing, isn't probably appropriate. Model #3 went for a direct "Will you be my Brie?" which is frankly quite confusing. An alternative may be "Vuoi essere la mia fontina d'ispirazione?", i.e. "Will you be my [little inspiration source | fontina]?" Probably not the best wordplay, but beats calling your significant other a slice of cheese...* |
| Mandarin | "Single's Day" should be translated as "单身日" instead of "光棍" because "光棍" is a slang term referring to single men, which is not polite. |

