# OpenReview forum: "Evaluating Cultural Alignment in Multilingual LLM Translations"
_EurIPS.cc/2025/Workshop/UPLB — Submitted to UPLB2025_

### Official Review · Reviewer_eo5D · 2025-10-17
**A Misfit for an ML Venue**

**Rating:** 2
**Confidence:** 5

**Review:**

This paper investigates how multilingual large language models (LLMs) translate and localize culturally nuanced figurative language, such as idioms and puns, across 20 languages and 24 dialects. Through human evaluation of 87 translations, the authors show that despite grammatical fluency, LLMs consistently fail to capture cultural resonance, highlighting limitations beyond data scale and linguistic proximity.

Strengths
- Practical focus: addresses a concrete, real-world challenge in multilingual LLM use -- translation of idiomatic and culturally loaded text -- using relatable marketing content.

Drawbacks
- Very limited sample size and statistical power: With only 87 evaluated items and fewer than three raters per language, the results have a very variance.
- This paper reads more like a statistical paper and does not use a typical template expected in an ML conference. I do not think this is a good fit for a machine learning workshop.

---

### Decision · Program_Chairs · 2025-11-03

Reject